# Phenotypic clines in herbivore resistance and reproductive traits in wild plants along an agricultural gradient

Hayley Schroeder[1]*, Heather Grab[2], Katja Poveda[1]

1 Department of Entomology, Cornell University, Ithaca, New York, United States of America, 2 School of Integrative Plant Sciences, Cornell University, Ithaca, New York, United States of America

* hayleyadair37@gmail.com

## Abstract

The conversion of natural landscapes to agriculture is a leading cause of biodiversity loss worldwide. While many studies examine how landscape modification affects species diversity, a trait-based approach can provide new insights into species responses to environmental change. Wild plants persisting in heavily modified landscapes provide a unique opportunity to examine species' responses to land use change. Trait expression within a community plays an important role in structuring species interactions, highlighting the potential implications of landscape mediated trait changes on ecosystem functioning. Here we test the effect of increasing agricultural landscape modification on defensive and reproductive traits in three commonly occurring Brassicaceae species to evaluate plant responses to landscape change. We collected seeds from populations at spatially separated sites with variation in surrounding agricultural land cover and grew them in a greenhouse common garden, measuring defensive traits through an herbivore no-choice bioassay as well as reproductive traits such as flower size and seed set. In two of the three species, plants originating from agriculturally dominant landscapes expressed a consistent reduction in flower size and herbivore leaf consumption. One species also showed reduced fitness associated with increasingly agricultural landscapes. These findings demonstrate that wild plants are responding to landscape modification, suggesting that the conversion of natural landscapes to agriculture has consequences for wild plant evolution.

## Introduction

Though habitat change is a natural part of global ecosystems, the scale and intensity of anthropogenic habitat disturbance in the last hundred years is unprecedented [1, 2]. In much of the world, human modified landscapes are frequently characterized by expansive monocultures with intensive chemical and mechanical management. Within these heavily modified landscapes, semi-natural habitat patches such as field margins and riparian buffers can function as havens for natural populations by creating connecting corridors and refuges [3–5]. While studies have documented declines in species diversity in agriculture dominated landscapes [6–8], species that persist in these environments may also experience shifts in diversity on the

**Data Availability Statement:** All 11 files are available from the Dryad Repository (DOI https://doi.org/10.5061/dryad.905qfttq6).

**Funding:** HS was supported by the Cornell University's Presidential Life Sciences Fellowship

and the National Science Foundation Graduate Research Fellowship Program under Grant No. DGE - 2139899/DGE - 1650441. Research was funded by the Cornell Entomology Griswold Endowment and the Atkinson Center Sustainable Biodiversity Fund. The funders had no role in study design, data collection and analysis, decision to publish, or preparation of the manuscript.

**Competing interests:** The authors have declared that no competing interests exist.

level of individual trait expression resulting in phenotypic clines across land use gradients. Examining intraspecific trait responses to landscape modification may provide further insight into species sensitivity to environmental change and help predict the consequences for ecosystem function [9].

There is growing evidence that trait shifts associated with phenotypic clines may contribute substantially to community trait variation, with significant implications for ecosystem processes [9–11]. In plants, intraspecific trait variation has been found to account for 32% of total trait variation among communities [12]. This variation can be attributed to a combination of heritable differences as well as plasticity [13] which collectively are an important mechanism allowing plants to track changing environmental conditions [14, 15]. Studies have found that wild plants experience fitness costs in landscapes dominated by agriculture [16, 17] raising questions about wild plants' capacity to track the unprecedented speed of anthropogenic environmental change.

There is already some evidence of phenotypic clines corresponding with human mediated landscape modification. Collectively these studies documented larger floral displays, reduced defenses, and advanced phenology in plants within highly modified landscapes [18–23]. Though these studies found similar patterns of phenotypic change, the predicted mechanism driving this change was not consistent in all studies. For example, both Moreira et al. (2019) and Thompson et al. (2016) documented a reduction in chemical defenses for plants in urbanized landscapes, however, the former was explained by variation in herbivore damage, while the latter was explained by colder ground temperatures in winter. Shifts in floral traits were consistently attributed to pollinator visitation [18, 20]. While these studies uncover important patterns and drivers of trait change associated with human mediated landscape modification, they are restricted to urbanization gradients, leaving patterns across agricultural gradients largely unexplored [24]. Given that agricultural landscapes cover over 40% of the earth surface [1, 25], this represents a critical gap in our understanding of plant responses to landscape modification.

Like urban landscapes, agricultural landscapes are characterized by unique biotic and abiotic conditions with the potential to alter trait distributions within local plant populations. For example, agricultural landscapes have been associated with a reduction in the abundance and diversity of pollinating insects [26, 27], soil degradation [28], and contamination by agrochemicals [29, 30]. A phenotypic cline would provide evidence that plants are responding to these novel environmental conditions. It is important to understand trait responses of plants growing in semi-natural spaces within agricultural landscapes due to the critical ecosystem services they provide by supplementing floral resources for pollinators [31, 32], creating reservoirs for natural enemies [33, 34], and connecting fragmented habitat patches [35, 36]. Therefore, trait changes have the potential not only to affect plant species persistence in modified landscapes, but also shape the community around them. Plant traits have been found to mediate intraspecific diversity in insects [37], highlighting the potential for shifts in intraspecific plant traits to have cascading effects on the broader community.

Though the direction and magnitude of insect responses to land use change can be highly variable by species [24], there are some generalizable patterns that we can use to formulate predictions for plant trait evolution across an agricultural gradient. Generally, land use change reduces the abundance and diversity of pollinators in a landscape [38–43]. Plants in landscapes with reduced pollinator abundances have been found to express both increased and decreased flower size [44, 45], suggesting that pollinator limitation may result in plant reproductive strategies that either increase or decrease reliance on pollinators [46]. For herbivores, there is evidence that increasing agricultural land area with low crop diversity increases specialist herbivore abundances that can utilize the dominant crop and decreases generalist herbivore

abundances [47–49]. Based on this evidence, we expect increasing agricultural landscape modification to reduce floral trait expression if plants are shifting away from a reproductive strategy reliant on pollinators. If the dominant herbivores in a landscape feed on non-crop plants as well, we expect to find an increase in resistance traits deterrent to herbivores. However, plants interact with pollinators and herbivores simultaneously, and the combined selective forces can be either opposing or synergistic [50–54].

In this study we test for evidence of a phenotypic cline across a gradient of agricultural landscape modification in three short-lived wild Brassicaceae: *Barbarea vulgaris* (Yellow Rocket), *Thlaspi arvense* (Field Pennycress) and *Capsella bursa-pastoris* (Shepherd's Purse). To evaluate trait changes in the absence of environmental variation, we collected seeds from plants growing in field margins along a gradient of increasing agricultural area in the surrounding landscape and grew their progeny together in a greenhouse common garden. Ultimately, this study seeks to examine wild plant responses in terms of herbivore resistance and reproductive traits to agricultural landscape modification, thus providing a foundation for formulating hypotheses and predictions about the mechanisms of landscape mediated trait change in future studies.

## Materials and methods

### Study system

While our three study species originate from Eurasia, they have a global distribution and have been present in North America for over a century [55]. These species provide a useful system to study landscape-mediated trait adaptation because they are abundant even in landscapes dominated by agriculture. Because they persist across the entire landscape gradient, these species are exposed to the full range of variation imposed by human-mediated landscape modification. All three species utilize mixed mating systems, but the potential for self-pollination varies by species, with *T. arvense* and *C. bursa-pastoris* producing a high proportion of self-fertilized seeds [55, 56], while *B. vulgaris* is primarily outcrossing [57]. Because all three species are considered naturalized in the landscape, no permits were required to collect seeds.

### Landscape analysis

Parent seeds were collected from sites in the Finger Lakes Region of New York State, USA in the fall 2019. Land use composition at each site was evaluated using the 2019 National Agricultural Statistics Service Cropland Data Layer for New York (USDA 2019) in QGIS 3.16 (QGIS Development Team, QGIS Geographic Information System). Within a 500, 1000, and 1500 m radius of each collection site we calculated the proportion categorized as agriculture, pasture, urban development, natural open and natural forested land area (see S1 Table for detailed land cover classifications). This a well-established method for evaluating the broad scale effects of landscape modification on species interactions and ecosystem function [58]. To confirm that the landscape composition has been consistent for at least the last decade, allowing for adaptation to take place across the gradient of landscape modification, we regressed agricultural landscape composition in 2008 against each subsequent year based on the USDA Cropland Data Layer (Fig 1).

### Seed collection from wild populations

At each site, parent plants were chosen haphazardly in semi-natural patches alongside cultivated crops (field margins, fallow fields, ditches). Sites were selected such that the existing variation in landscape composition at the broader scale from 500–1500m around each collection

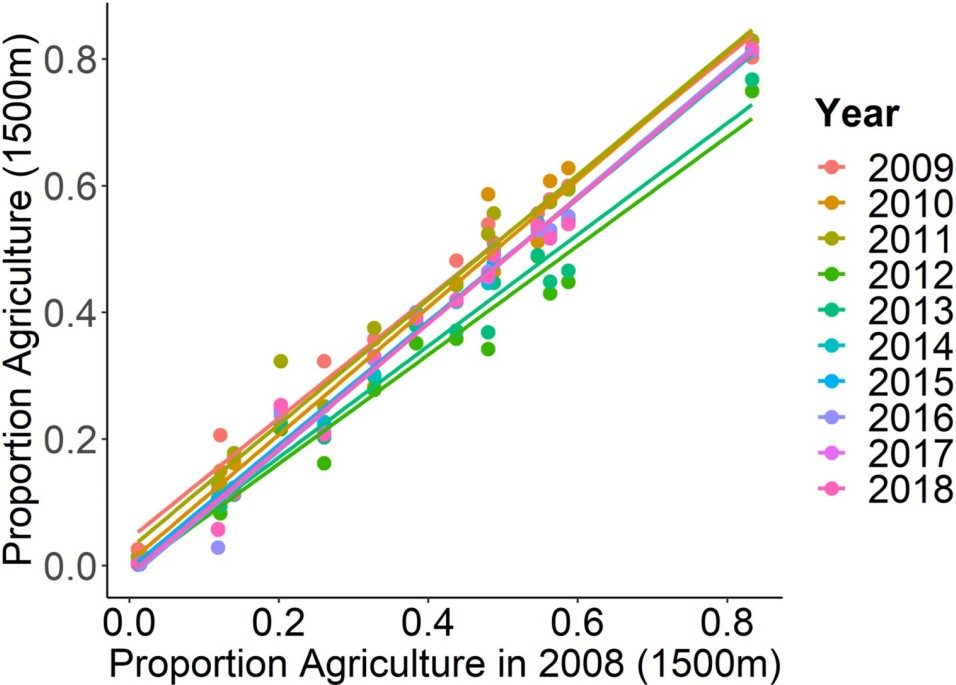

**Fig 1. Landscape composition has been consistent for the last decade.** Correlation between the USDA Cropland Data Layer landscape classification for agricultural land cover at 1500m in 2008 and each subsequent year through 2018 for each collection site. Each point represents a collection site, and each line represents the landscape composition across all sites in a different year compared to 2008.

site represented a gradient with an increasing proportion of agricultural land cover (Fig 1). Seeds of *B. vulgaris* were collected at 13 sites from at least five parent plants at all sites except one where only four parent plants were present. Seed of *T. arvense* were collected from 15 sites, with at least five parent plants at all but two sites where only three parent plants were present. For *C. bursa-pastoris*, seeds were collected at 16 sites, with at least five parent plants at all sites except two where only four parent plants were present. Seeds were allowed to fully dry on the plant before collection and were kept separated by parent and stored in the freezer for at least a month to promote germination.

## Germination and propagation

For each species, twenty-five seeds from each of five randomly selected parents (or the maximum number of parents available when less than five parents were found at a site) from each collection site were weighed together to the nearest .01mg to test for differences in seed mass from the field. Seeds were then planted in trays with Cornell Mix (see soil preparation in Supporting Information) in a greenhouse at approximately 20˚ C with a 12-hour photoperiod and biweekly fertilization with Jack's Professional® water-soluble fertilizer 21-5-20 as 300 PPM (JR Peters, Inc., Allentown, PA USA). Total germination was recorded after 14 days. Seeds germinated from all *B. vulgaris* parent plants, while eight out of 71 *T. arvense* parent plants and 35 of 72 *C. bursa-pastoris* parent plants resulted in no germination. Replanting *C. bursa-pastoris* seeds to account for the low germination resulted in 1–25 germinated seedlings per site. Five randomly selected seedlings per parent (or the maximum number germinated when less than five seedlings from one parent germinated) were transplanted into individual pots (4 x 4 x 3.5

in) with Cornell Mix resulting in 320 *B. vulgaris*, 282 *T. arvense*, and 231 *C. bursa-pastoris* seedlings. The seedlings were arranged randomly in the greenhouse to account for any possible variation in light, temperature, or watering.

As biennials, it is common for brassica species to require a cold period to initiate flowering. All but two *C. bursa-pastoris* plants and 72 *T. arvense* plants bolted without vernalization. The remaining 210 *T. arvense* plants and all 320 *B. vulgaris* plants were moved into a cold growth chamber to vernalize. The temperature was set to 4° C with a 10-hour photoperiod (corresponding to median winter day length) and the plants were watered only as needed to prevent wilting. Plants were moved back into the greenhouse after approximately 4 months. Out of 319 *B. vulgaris* plants, 308 individuals successfully bloomed after vernalization. All but 8 of the 282 individual *T. arvense* and 2 of the 231 *C. bursa-pastoris* plants successfully bloomed.

### Floral traits

For all species, we removed one of the first 5 flowers that opened on a plant. All petals were removed and fixed to a piece of paper with transparent tape. Petal length and width were measured using a microscope at 12.5 magnification and CellSens software (Olympus SZX10 stereo microscope using the digital measurement tool in the cellSens software (Olympus Corp. Tokyo, Japan)) to the nearest hundredth mm. Anther-stigma distance was evaluated by measuring the length of the stamen and pistil and then subtracting the two values (stamen-pistil). The longest petal was used to measure petal length and width. To estimate petal area, we used the formula for an oval (½ length x ½ width x π). We evaluated self-pollination once the plants had finished blooming by recording the proportion of fertile seed pods out of 30 on three stalks (for a total of 90 per plant) when counting from the base. A seed pod was classified as fertile if at least one seed was developing inside. Plants were spaced in the greenhouse so that no individuals were touching, to prevent any neighboring plants from exchanging pollen.

### Plant harvest/seed collection

For *C. bursa-pastoris* and *T. arvense*, once seed pods had dried but not shattered, we collected the first 15 unshattered seed pods from one of the most central stalks. After the seeds were collected, plants were cut back to the soil surface and placed in paper bags. The bags were left in the greenhouse to dry and moved into a drying oven for 48 hours at approximately 38° C. The plants were weighed immediately after they were removed from the drying oven. The seed pods that were collected from the plants were weighed separately. For *C. bursa-pastoris*, seeds and seed pods were weighed together due to difficulty separating the small seeds from shattered pod debris. For *T. arvense*, the seeds were separated easily from the seed pods, and therefore only the seeds themselves were weighed. Due to the COVID19 pandemic related campus closures, *B. vulgaris* seeds were collected before the pods were ripe. The green pods were dried in a drying oven for 3 days before the seeds and seed pods were weighed together. This method was validated using a subset of *B. vulgaris* seed pods from a separate experiment confirming that seed pod mass and seed mass are highly correlated (Pearson's r = 0.90, $P < 0.001$).

### Bioassay

We conducted no choice bioassays using the generalist herbivore, *Trichoplusia ni*. Bioassays were conducted independently for each species following the same protocol. Seeds from the same parent plants used to evaluate reproductive traits were sown in a growth chamber with a temperature of 21° C, 15 h fluorescent light per day and daily watering. Once the seedlings had produced at least four true leaves, we randomly selected and transferred two seedlings from each parent to individual pots (4 x 4 x 3.5 in) with Cornell Mix and randomized the position of

the pots among rows and columns within the growth chamber. Approximately 2–4 weeks after transplanting (depending on the species) two leaves were removed from each plant (126 *B. vulgaris*, 150 *C. bursa-pastoris*, and 124 *T. arvense* plants) starting with the first fully expanded leaf and continuing around the rosette. Each leaf was placed in an individual florist tube with water and enclosed in a plastic cup to contain the caterpillar with the leaf. Each leaf received one caterpillar that had fed on Southland Products Cabbage Looper Diet for two days to maximize caterpillar survival. Caterpillars were weighed immediately prior to placement and remained on the leaves for 3 days in a growth chamber with a temperature of 25˚ C and 15 h of fluorescent light per day. On the third day, we weighed the caterpillars to the nearest .01mg and measured the total leaf area and leaf area consumed using LeafByte [59].

## Statistics

We used R programming software (v.4.1.0) for statistical analyses [60]. We first computed Pearson correlation coefficients evaluating how the landscape composition across sites for each year correlated with the composition in the earliest year available (2008) to confirm the landscape composition has been stable across the last decade. Given the high number of landscape variables and scales (S1 Table), we then conducted a principal component analysis combining the landscape classifications found to be the most predictive for each individual species across scales to produce PC1 and PC2 values that summarized the relative composition of each source site (Fig 2). The PC1 value represents increasing agricultural land cover, explaining 47.6% of the variation in landscape composition, and was used for all species in all further analyses. We consider these results to be highly conservative given that we found each species showed the strongest response to a unique landscape type and scale when analyzed independently (S1–S3 Figs; S3–S5 Tables). We present the results from the principal component landscape variable here to uncover generalizable patterns of trait change associated with agriculture.

To evaluate changes in trait expression across the land use gradient we fit separate linear mixed models using the 'lme4' R package with each trait as the response variable and the PC1 landscape variable described in the previous analysis as a fixed effect [61]. We fit individual linear mixed-effects models for each species and trait combination (field collected seed mass, germination rate, petal area, anther-stigma distance, plant mass, self-pollination rate, self-pollination seed mass, caterpillar consumption efficiency, leaf area consumed, and caterpillar relative growth rate). Each model included a random effect of parent plant to control for multiple offspring from each parent which was nested within collection site to control for non-independence of parents within a site. Models for the parent-level traits of field collected seed mass and germination rate only included collection site as a random effect. For models evaluating petal size, we included plant mass as a predictor to control for variation in plant size. In each model evaluating *T. arvense* reproductive traits (petal area, seed mass, plant mass) we included vernalization as a random effect. We also ran a logistic regression to test the extent that the landscape gradient explains the probability that a *T. arvense* plant required vernalization to bloom.

To evaluate defensive traits, we used a logistic regression to test if caterpillar survival correlated significantly with the landscape gradient. Given there was low mortality that did not correlate with the landscape gradient, we excluded any caterpillars that did not survive to the end of the experiment in all future models. In the models evaluating defensive traits from the bioassay (caterpillar consumption efficiency, leaf area consumed, and caterpillar relative growth rate), we included bolting status (a binary variable describing whether a plant had developed reproductive structures as the time of the experiment) as a random effect for *T. arvense* and *C.*

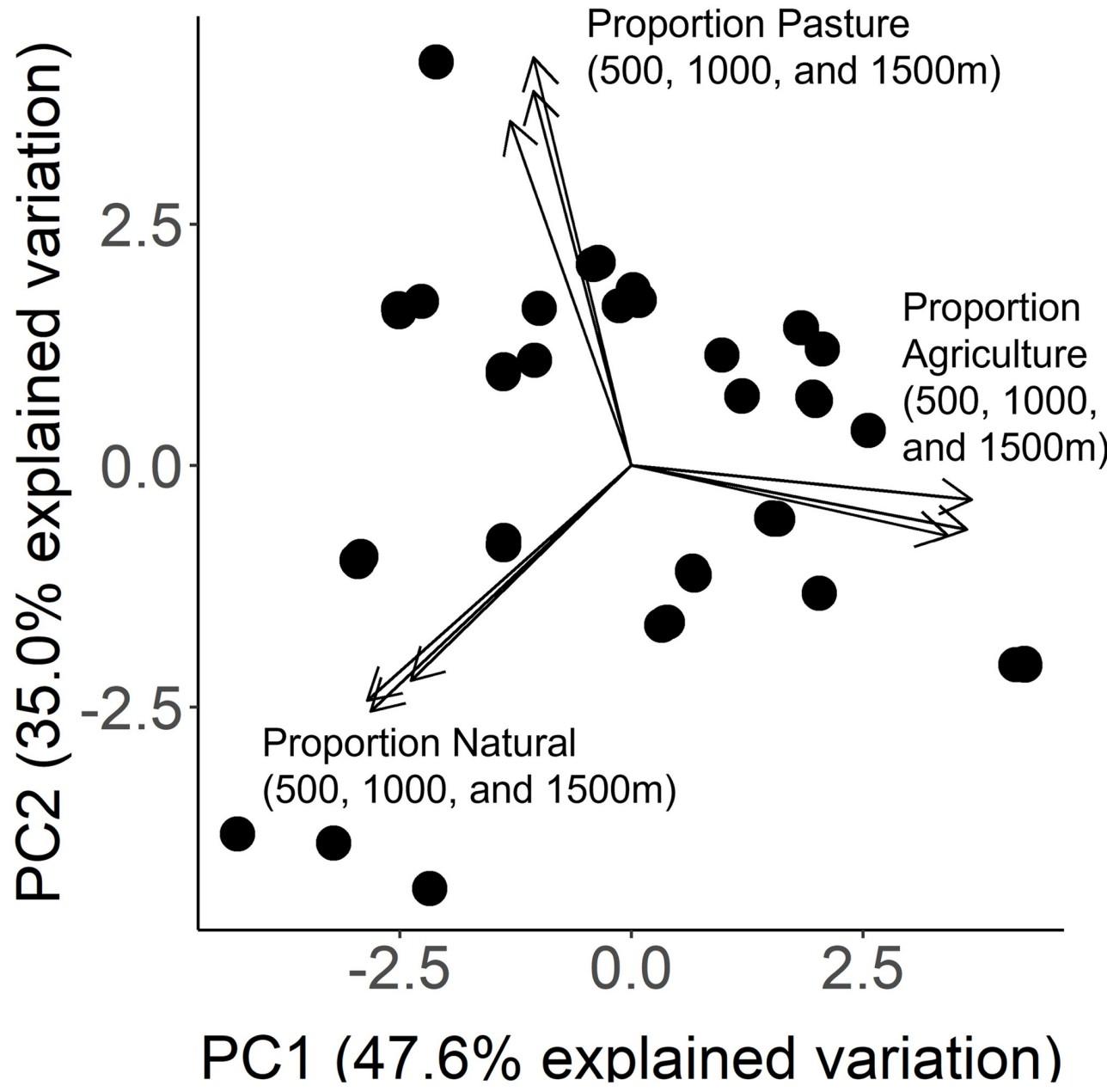

**Fig 2. Principal component analysis summarizing landscape composition across sites.** A principal component analysis combining the most predictive landscape types (agriculture, pasture, and unforested natural land covers) for all study species across three scales surrounding the collection site (500, 1000 and 1500m). Each point represents a site where seeds were collected. The first two principal components account for at least 82% of the variation in landscape composition. PC1 has positive and negative loadings reflecting agricultural land cover, and PC2 has positive and negative loadings reflecting pastural land cover.

*bursa-pastoris* to account for variation in plant ontogeny. For all models, we evaluated the significance of individual trait responses using a type three ANOVA. Mantel tests indicated spatial autocorrelation in the residuals of the model testing self-pollination rate in *B. vulgaris*, and the site coordinates were therefore included as fixed effects in this model (Dray & Dufour, 2007 [62]; S2 Table). There was no significant spatial autocorrelation in the residuals for all other models (S2 Table).

## Results

### Landscape stability and field collected data

There was a strong positive correlation between the landscape cover across sites in 2008 and the landscape composition for each subsequent year, providing evidence for landscape stability in the last decade (r > 0.98, $P$ < 0.0001). Of the seeds collected from the field, neither the seed mass nor germination rates correlated significantly with the land use gradient except for C. *bursa-pastoris*, where a lower germination rate was found for seeds collected from sites with greater agricultural habitat modification ($\chi^2$ = 6.7344, $P$ < 0.01, Fig 3A–3C, Table 1).

Results of individual linear mixed-effects models investigating the effects of agricultural landscape modification (PC1) on field collected seed mass and germination rate in three wild brassica species. Agricultural landscape modification refers to the extent of agricultural land cover in a radius surrounding the collection site. Collection site was included as a random effect in all models. Statistically significant predictors are indicated in bold ($P$ < 0.05).

### Reproductive traits

For both *B. vulgaris* and *C. bursa-pastoris*, petal area decreased with increasing agricultural landscape modification, while *T. arvense* petal size did not change (*B. vulgaris*: $\chi^2$ = 10.53, $P$ < 0.01, *C. bursa-pastoris*: $\chi^2$ = 4.34, $P$ < 0.01, Fig 3D–3F, Table 2). However, for *T. arvense*, plants from landscapes with higher agricultural landscape modification were more likely to require vernalization to bloom than plants from more natural landscapes ($\chi^2$ = 38.08, $P$ < 0.0001).

We found no significant relationship for any species between stigma-anther distance, plant height, plant mass, or auto-fertility and agricultural landscape modification. In *C. bursa-pastoris*, self-pollinated seed mass declined significantly with increasing agricultural land cover (Table 2, Fig 3G). There was a marginal negative correlation for *B. vulgaris* self-pollination rate and a marginal positive correlation for *T. arvense* seed mass with increasing agricultural landscape modification (*B. vulgaris*: $\chi^2$ = 366, $P$ = 0.056, *T. arvense*: $\chi^2$ = 3.46, $P$ = 0.063, Table 2, Fig 3H and 3I).

Results of general linear mixed models investigating the effects of agricultural landscape modification (PC1) on petal size, stigma-anther distance, plant mass, auto-fertility, and mass of self-fertilized seeds in three wild brassica species. Agricultural landscape modification refers to the extent of agricultural land cover in a radius surrounding the collection site. Collection site and parent plant were included as a random effect in all models. The site coordinates are included in the model of self-pollination rate in *B. vulgaris* to account for spatial autocorrelation in the residuals of the initial model. For *T. arvense*, vernalization status was also included as a random effect to account for the proportion of plants that underwent a cold treatment to initiate bloom. Statistically significant predictors ($P$ < 0.05) are indicated in bold and marginal predictors ($P$ < 0.1) are italicized.

### Herbivore resistance assessment

In the herbivore no-choice bioassay, 96.8% of 252 *T. ni* caterpillars survived on *B. vulgaris*, 88.7% survived out of 248 on *T. arvense*, and 97.3% survived out of 300 on *C. bursa-pastoris*. The likelihood of caterpillar mortality did not correlate with increasing agricultural land cover (*B. vulgaris*: $\chi^2$ = 0.135, $P$ = 0.713; *C. bursa-pastoris*: $\chi^2$ = 0.273, $P$ = 0.601; *T. arvense*: $\chi^2$ = 1.229, $P$ = 0.268). For both *B. vulgaris* and *C. bursa-pastoris*, caterpillars consumed less leaf area on plants from landscapes with greater agricultural land cover (Table 3, Fig 3J–3L). For all species, the leaf area and *T. ni* initial mass were positively correlated with consumed leaf area

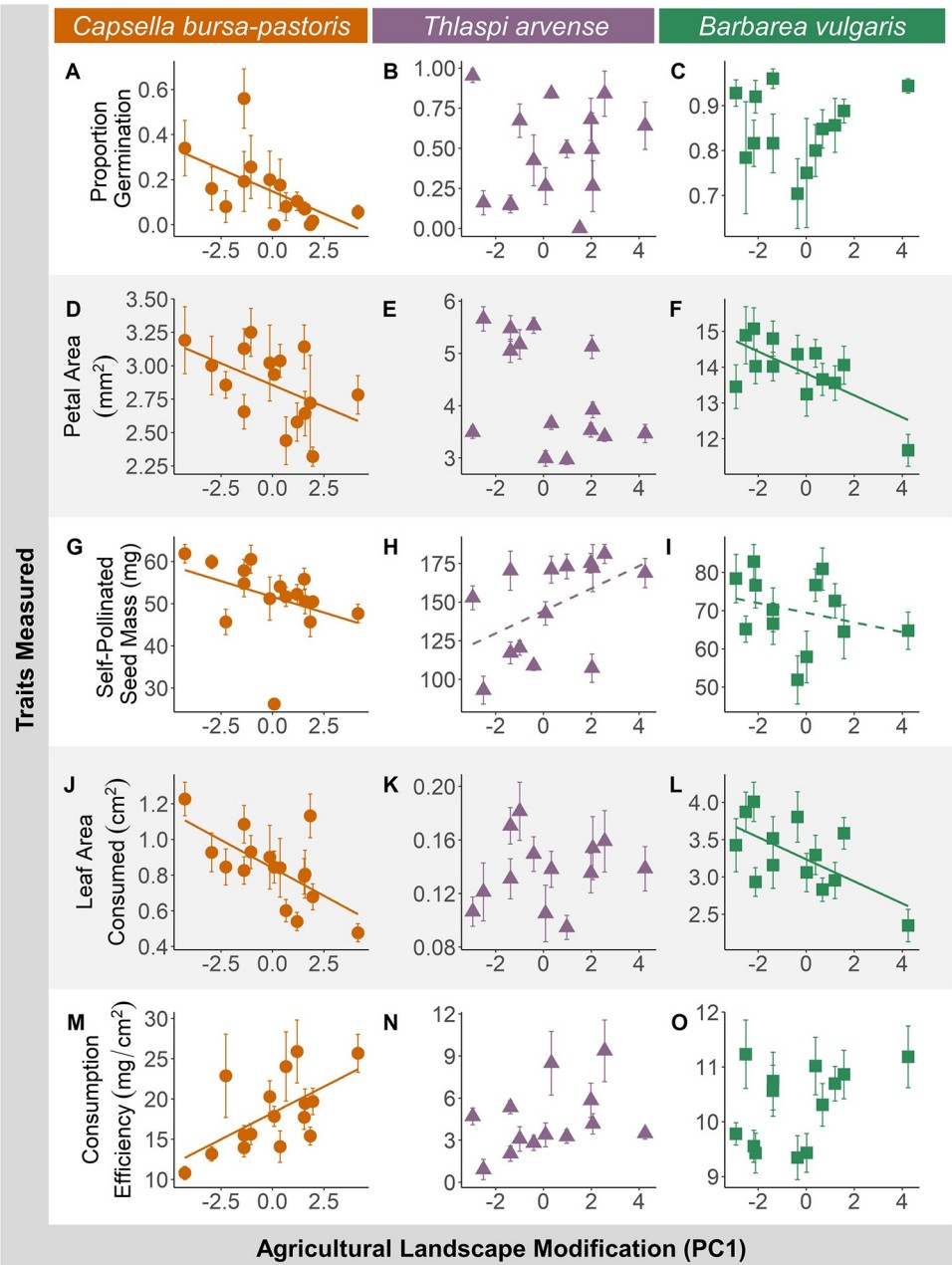

**Fig 3. Plant responses to increasing agricultural landscape modification vary by species and trait.** Columns represent individual species, and each row displays one of the traits evaluated. For the seeds collected from the field, we measured the proportion of seeds that germinated out of 25 total seeds (A, B, C). In the greenhouse plants were allowed to bloom and set seed to measure petal size (D, E, F) and self-pollinated seed mass (G, H, I). In an herbivore no-choice bioassay with *Trichoplusia ni* we measured the leaf area consumed by caterpillars (J, K, L) and caterpillar consumption efficiency (M,N,O). Agricultural landscape modification (PC1) refers to PC1 values from a principal component analysis summarizing pasture, natural, and agricultural land cover at three scales (500, 1000, and 1500m) for each site. Higher values indicate a greater proportion agriculture in the surrounding landscape and lower values indicate a greater proportion open (non-forested) natural area. Solid regression lines indicated significant regression coefficients ($P < 0.05$), dashed lines indicate marginal regression coefficients ($P < 0.1$), and no regression line indicates no relationship. Error bars represent standard error.

**Table 1. Analysis of landscape effects on seed germination rate for three species.**

| Species | Response variable | N Parents | N Sites | Predictor | Chisq | Df | p-value |
|---------|-------------------|-----------|---------|-----------|-------|----|---------| 
| *B. vulgaris* | Field Seed Mass | 63 | 13 | PC1 | 1.0696 | 1 | 0.301 |
| | Germination Rate | 63 | 13 | PC1 | 0.2264 | 1 | 0.6342 |
| *C. bursa-pastoris* | Field Seed Mass | 78 | 16 | PC1 | 0.0085 | 1 | 0.9264 |
| | **Germination Rate** | **78** | **16** | **PC1** | **6.7344** | **1** | **0.0095** |
| *T. arvense* | Field Seed Mass | 71 | 15 | PC1 | 0.9013 | 1 | 0.3424 |
| | Germination Rate | 71 | 15 | PC1 | 0.2911 | 1 | 0.5895 |

(Table 3). Leaf area was positively correlated with caterpillar relative growth rate for all species, while agricultural land cover had no effect (Table 3). Caterpillar consumption efficiency was significantly positively correlated with agricultural land cover in *C. bursa-pastoris*, but not in *B. vulgaris* and *T. arvense* (Table 3, Fig 3M–3O). Caterpillar initial mass was a significant positive predictor of consumption efficiency for *B. vulgaris* and negative predictor for *T. arvense*. Leaf area was a significant negative predictor of consumption efficiency for *C. bursa-pastoris*.

Results of general linear mixed models investigating the effects of agricultural landscape modification (PC1) on caterpillar leaf area consumption, relative growth rate, and consumption efficiency (mass gained per unit area eaten) in a no choice bioassay with a generalist herbivore (*Trichoplusia ni*) feeding on three wild brassica species. Agricultural landscape modification refers to the extent of agricultural land cover in a radius surrounding the collection site. Collection site and parent plant were included as a random effect in all models. For *C. bursa-pastoris* and *T. arvense*, bolting status was also included as a random effect as a significant proportion of the plants began to bloom at the onset of the experiment. Statistically significant predictors (*P* < 0.05) are indicated in bold and marginal predictors (*P* < 0.1) are italicized.

**Table 2. Analysis of landscape effects on plant reproductive traits.**

| Species | Response variable | N plants | N Sites | Predictor | Chisq | Df | p-value |
|---------|-------------------|----------|---------|-----------|-------|----|---------| 
| *B. vulgaris* | **Petal Area** | **309** | **13** | **PC1** | **10.531** | **1** | **0.0012** |
| | | | | Plant Mass | 0.0093 | 1 | 0.9234 |
| | Stigma-Anther Distance | 312 | 13 | PC1 | 0.0006 | 1 | 0.9803 |
| | Plant Mass | 311 | 13 | PC1 | 0.1542 | 1 | 0.6946 |
| | *Self-Pollination Rate* | *305* | *13* | *PC1* | *3.6598* | *1* | *0.0557* |
| | | | | **X Coordinate** | **8.2280** | **1** | **0.0041** |
| | | | | Y Coordinate | 0.3644 | 1 | 0.5461 |
| | Self-Pollinated Seed Mass | 307 | 13 | PC1 | 0.9705 | 1 | 0.3246 |
| *C. bursa-pastoris* | **Petal Area** | **196** | **16** | **PC1** | **4.3404** | **1** | **0.0372** |
| | | | | *Plant Mass* | *3.0199* | *1* | *0.0823* |
| | Stigma-Anther Distance | 216 | 16 | PC1 | 1.6765 | 1 | 0.1954 |
| | Plant Mass | 209 | 16 | PC1 | 0.9104 | 1 | 0.34 |
| | Self-Pollination Rate | 218 | 16 | PC1 | 0.8209 | 1 | 0.3649 |
| | **Self-Pollinated Seed Mass** | **201** | **16** | **PC1** | **9.5532** | **1** | **0.0020** |
| *T. arvense* | Petal Area | 250 | 14 | PC1 | 1.5116 | 1 | 0.2189 |
| | | | | Plant Mass | 0.3114 | 1 | 0.5768 |
| | Stigma-Anther Distance | 256 | 14 | PC1 | 0.2458 | 1 | 0.6201 |
| | Plant Mass | 269 | 14 | PC1 | 0.1555 | 1 | 0.6933 |
| | Self-Pollination Rate | 269 | 14 | PC1 | 0.0216 | 1 | 0.8831 |
| | *Self-Pollinated Seed Mass* | *241* | *14* | *PC1* | *3.4603* | *1* | *0.0629* |

**Table 3. Landscape affects on plant defensive traits.**

| Species | Response variable | N Plants | N Sites | Predictor | Chisq | Df | p-value |
|---|---|---|---|---|---|---|---|
| *B. vulgaris* | **Leaf Area Consumed** | **243** | **13** | **PC1** | **6.8196** | **1** | **0.00902** |
| | | | | **Leaf Size** | **6.9180** | **1** | **0.00853** |
| | | | | ***T. ni* Initial Mass** | **57.461** | **1** | **<0.0001** |
| | Relative Growth Rate | 243 | 13 | PC1 | 2.6212 | 1 | 0.10544 |
| | | | | **Leaf Size** | **3.8853** | **1** | **0.04871** |
| | Consumption Efficiency | 243 | 13 | PC1 | 2.3731 | 1 | 0.12344 |
| | | | | *Leaf Size* | *2.9017* | *1* | *0.08848* |
| | | | | ***T. ni* Initial Mass** | **5.4733** | **1** | **0.01931** |
| *C. bursa-pastoris* | **Leaf Area Consumed** | **288** | **16** | **PC1** | **5.2980** | **1** | **0.02135** |
| | | | | **Leaf Size** | **28.020** | **1** | **<0.0001** |
| | | | | ***T. ni* Initial Mass** | **30.515** | **1** | **<0.0001** |
| | Relative Growth Rate | 288 | 16 | PC1 | 0.0841 | 1 | 0.77175 |
| | | | | **Leaf Size** | **6.8043** | **1** | **0.00909** |
| | **Consumption Efficiency** | **288** | **16** | **PC1** | **3.9268** | **1** | **0.04752** |
| | | | | **Leaf Size** | **19.321** | **1** | **<0.0001** |
| | | | | *T. ni* Initial Mass | 2.6932 | 1 | 0.10078 |
| *T. arvense* | Leaf Area Consumed | 219 | 13 | PC1 | 0.1570 | 1 | 0.69192 |
| | | | | **Leaf Size** | **77.594** | **1** | **<0.0001** |
| | | | | ***T. ni* Initial Mass** | **4.2802** | **1** | **0.03856** |
| | Relative Growth Rate | 216 | 13 | PC1 | 1.1599 | 1 | 0.2815 |
| | | | | **Leaf Size** | **25.276** | **1** | **<0.0001** |
| | Consumption Efficiency | 217 | 13 | PC1 | 1.8038 | 1 | 0.17925 |
| | | | | Leaf Size | 0.5398 | 1 | 0.46252 |
| | | | | ***T. ni* Initial Mass** | **4.7966** | **1** | **0.02852** |

## Discussion

The consequences of agricultural landscape modification on biodiversity and ecosystem function are at the forefront of ecological research [63–65]. However, the cascading effects of this landscape modification on individual trait responses have, until now, been largely unexplored. Recently, Mitchell et al. (2021) [66] found that proximity to cultivated sunflowers resulted in homogenized selection on floral traits for wild sunflowers. Here, we build on these findings by documenting phenotypic clines in both defensive and reproductive traits across three related plant species. These landscape mediated changes in plant phenotypes have the potential to shape the trait diversity of the broader community [37].

Our findings document a reduction in floral display size in plants from agriculture dominant landscapes for two of the three species examined. This pattern is counter to the findings along urbanization gradients documenting increased floral display size associated with urban environments [18, 20], highlighting the unique conditions imposed by each landscape type. Mitchell et al. (2021) [66] found evidence of selection for higher ray length in wild sunflowers further from cultivated sunflower, aligning with our findings here in Brassicaceae. There are a number of potential biotic and abiotic mechanisms that may be mediating these changes in floral traits [67], however, insect pollinators are frequently identified as key drivers. For example, Brys and Jacquemyn (2012) found that pollinator depleted environments are associated with reduced flower size and increased autonomous self-pollination. There is strong evidence that agricultural landscapes support a reduced abundance and diversity of pollinators [26], making this mechanism a priority for exploration in future studies. A lack of response by the

third species (*T. arvense*) may be explained by the fact that in the individual analyses, trait variation was best predicted by the proportion pasture in the landscape rather than agriculture. However, it may also indicate resilience or that this species is responding by varying another trait not measured in this study such as floral rewards or number of open flowers. Future studies should evaluate how variation in different floral traits affect total plant fitness and whether this is predicted by insect interactions or abiotic factors.

We also found that the land use gradient predicted metrics of plant defense for two species. *T. ni* caterpillars consumed less leaf area and developed more efficiently on plants from agriculturally dominant landscapes. Given that all plants were grown under equivalent conditions, these observed differences in consumption and assimilation are likely due to increased plant defenses in agricultural landscape, but further work is necessary to confirm this mechanism. There is a large body of evidence that herbivore pressure and plant defenses vary predictably along environmental gradients such as elevation, latitude, or plant diversity [68–71]. Once again, studies examining plant defenses in urban landscapes have found the opposite pattern to what we observed here, demonstrating a decrease in plant defenses within modified urban landscapes [21, 72]. The few studies that have tested these patterns of herbivory in wild plants growing alongside agriculture have shown mixed results, where some report reduced herbivore pressure associated with agriculture [73], while others document augmented herbivore damage on wild plants [74]. There are a number of possible factors contributing to these varied results, including the natural history of the major insect herbivore, the relative attractiveness of adjacent crops compared to the wild plant community, potential insecticide spillover onto wild plants in intensively managed landscapes, and other abiotic factors like temperature [23, 48, 75–77].

We examined metrics of fitness through seed mass and germination rate in the wild parent plants and through seed mass and self-fertilization rate in the common garden plants. Of the field collected data, only *C. bursa-pastoris* demonstrated a significant decline in seed germination in agriculture dominant landscapes. This is consistent with existing studies comparing wild plant reproduction across agricultural gradients [16, 78] and may also be the result of a dormancy mechanism adaptive in high disturbance agricultural environments [79]. Of the common garden plants, again *C. bursa-pastoris* was the only species that expressed a fitness cost with plants from agriculture dominant landscapes producing lighter seeds through self-fertilization. The combination of low germination rate in seeds from the parent plants and low seed weight in the common garden seeds may indicate inbreeding depression [80], potentially due to population fragmentation associated with homogenized landscapes and reduced pollinator availability to connect isolated populations. The absence of a fitness cost for the other two species could provide evidence of adaptation to the local environment in agricultural landscapes.

In this study we document phenotypic clines across a landscape modification gradient, however, with these data we cannot disentangle the relative degree that plant phenotypic differences can be explained by genetic differences, plasticity, and maternal effects. Given the importance of plasticity and maternal effects in enabling plants to track rapidly changing environmental conditions [81], examining overall trait responses provides a more complete estimate of wild plant responses to agricultural landscape modification. Future studies should test the relative contribution of plasticity and heritable change and examine whether these changes in trait values are adaptive. These studies should evaluate how different plant phenotypes perform across the landscape to test for evidence of selection and document insect interactions and abiotic conditions to identify the mechanisms driving plant trait change.

To date, much of the literature examining the consequences of agricultural landscape modification on plant communities has focused primarily on changes in species diversity. Here we demonstrate the potential for intraspecific trait change in species that persist within highly

modified landscapes, highlighting the need to incorporate trait-based analyses into landscape scale studies. Given that wild plant communities play a critical role in habitat connectivity and stability in highly fragmented, human altered landscapes, understanding how they respond to land use change can inform predictions about the responses and resilience of entire communities [37, 82]. We hope this work will spur further research examining plant trait change across species and land use types to expose broader patterns of plant trait change in human modified landscapes.

## Conclusions

Across three Brassicaceae species, we found evidence for trait changes associated with increasing agricultural landscape modification in two species. For these species, plants originating from sites with greater agricultural modification in the surrounding landscape produced smaller flowers and received less damage in an herbivore no-choice bioassay than plant originating from landscapes with greater natural land cover. Because pollinators often impose selection on larger floral traits and herbivores on increased resistance, these findings may suggest greater pollinator mediated selection in natural landscapes and greater herbivore mediated selection in agricultural landscapes. However, because the outcome of selection is ultimately a result of the combined contribution of pollinators and herbivores, future studies should evaluate the relative role that pollinators and herbivores play in driving landscape mediated changes in plant traits.

## Supporting information

**S1 File. Description of statistical analysis for all supporting documents.**
(DOCX)

**S2 File. Soil components and preparation for Cornell Mix soil used in experiments.**
(DOCX)

**S1 Fig.** Germination rate (A) and seed mass (B) of seeds collected from *B. vulgaris* plants growing at sites representing a gradient of increasing open (non-forested) natural land cover in a 1000m radius around the collection site. Petal area (C), plant mass (D), self-pollinated seed mass (E), percent aborted seed pods (F), and stigma-anther distance (G) of the offspring grown from the parent populations in a greenhouse common garden. Caterpillar consumed leaf area (H) and consumption efficiency (I) from a no-choice herbivore bioassay represent a proxy for plant defense traits in the offspring grown from the parent populations. Solid regression lines indicate significant correlations, dashed indicate marginal correlations, and no regression line indicates no relationship. Error bars represent standard error.
(DOCX)

**S2 Fig.** Germination rate (A) and seed mass (B) of seeds collected from *C. bursa-pastoris* plants growing at sites representing a gradient of increasing agricultural land cover in a 1500m radius around the collection site. Petal area (C), plant mass (D), self-pollinated seed mass (E), percent aborted seed pods (F), and stigma-anther distance (G) of the offspring grown from the parent populations in a greenhouse common garden. Caterpillar consumed leaf area (H) and consumption efficiency (I) from a no-choice herbivore bioassay represent a proxy for plant defense traits in the offspring grown from the parent populations. Solid regression lines indicate significant correlations, dashed indicate marginal correlations, and no regression line indicates no relationship. Error bars represent standard error.
(DOCX)

**S3 Fig.** Germination rate (A) and seed mass (B) of seeds collected from *T. arvense* plants growing at sites representing a gradient of increasing pasture land cover in a 500m radius around the collection site. Petal area (C), plant mass (D), self-pollinated seed mass (E), percent aborted seed pods (F), and stigma-anther distance (G) of the offspring grown from the parent populations in a greenhouse common garden. Caterpillar consumed leaf area (H) and consumption efficiency (I) from a no-choice herbivore bioassay represent a proxy for plant defense traits in the offspring grown from the parent populations. Solid regression lines indicate significant correlations, dashed indicate marginal correlations, and no regression line indicates no relationship. Error bars represent standard error.
(DOCX)

**S1 Table. CDL land cover types included within the broad land use classifications used in the PCA in the main text and individual analyses included in the supplement.**
(DOCX)

**S2 Table. Results of Mantel test of spatial autocorrelation in the residuals of the final models.** Statistically significant predictors ($P < 0.05$) are indicated in bold.
(DOCX)

**S3 Table. Results of general linear mixed models investigating the effects of open natural land cover on all measured traits for *B. vulgaris*.** Parent plant nested within collection site was included as a random effect in all models. Statistically significant predictors ($P < 0.05$) are indicated in bold and marginal predictors ($P < 0.1$) are italicized. *Trichoplusia ni* caterpillars used in the leaf bioassay are included in the predictor column as *T.ni*.
(DOCX)

**S4 Table. Results of general linear mixed models investigating the effects of agricultural land cover on all measured traits for *C. bursa-pastoris*.** Parent plant nested within collection site was included as a random effect in all models. Statistically significant predictors ($P < 0.05$) are indicated in bold and marginal predictors ($P < 0.1$) are italicized. *Trichoplusia ni* caterpillars used in the leaf bioassay are included in the predictor column as *T.ni*.
(DOCX)

**S5 Table. Results of general linear mixed models investigating the effects of pasture land cover on all measured traits for *C. bursa-pastoris*.** Parent plant nested within collection site was included as a random effect in all models. Statistically significant predictors ($P < 0.05$) are indicated in bold and marginal predictors ($P < 0.1$) are italicized. *Trichoplusia ni* caterpillars used in the leaf bioassay are included in the predictor column as *T.ni*.
(DOCX)

## Acknowledgments

We thank Anurag Agrawal, Monica Geber, and Andre Kessler for suggestions regarding project design and comments on earlier drafts of the manuscript. We thank Annika Salzberg, Casey Hale, and Emma Harte for assistance with plant care and data collection and Erika Mudrak at the Cornell Statistical Consulting Center for statistical advice.

## Author Contributions

**Conceptualization:** Heather Grab, Katja Poveda.

**Data curation:** Hayley Schroeder.

**Formal analysis:** Hayley Schroeder.

**Funding acquisition:** Hayley Schroeder.

**Investigation:** Hayley Schroeder.

**Methodology:** Hayley Schroeder.

**Supervision:** Katja Poveda.

**Visualization:** Hayley Schroeder.

**Writing – original draft:** Hayley Schroeder.

**Writing – review & editing:** Heather Grab, Katja Poveda.

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
