## [Decision Letter · Decision Letter 0]

20 Feb 2023

PONE-D-23-01036Phenotypic clines in herbivore resistance and reproductive traits in wild plants along an agricultural gradientPLOS ONE

Dear Dr. Schroeder,

Thank you for submitting your manuscript to PLOS ONE. After careful consideration, we feel that it has merit but does not fully meet PLOS ONE’s publication criteria as it currently stands. Therefore, we invite you to submit a revised version of the manuscript that addresses the points raised during the review process.

We look forward to receiving your revised manuscript.

Kind regards,

Andrea Mastinu

Academic Editor

PLOS ONE

Journal Requirements:

Reviewers' comments:

Reviewer's Responses to Questions

**Comments to the Author**

1. Is the manuscript technically sound, and do the data support the conclusions?

Reviewer #1: Yes

Reviewer #2: Yes

2. Has the statistical analysis been performed appropriately and rigorously? 

Reviewer #1: Yes

Reviewer #2: Yes

3. Have the authors made all data underlying the findings in their manuscript fully available?

Reviewer #1: Yes

Reviewer #2: Yes

4. Is the manuscript presented in an intelligible fashion and written in standard English?

Reviewer #1: Yes

Reviewer #2: Yes

5. Review Comments to the Author

Reviewer #1: I have carefully reviewed the manuscript titled "Phenotypic clines in herbivore resistance and reproductive traits in wild plants along an agricultural gradient," authored by Schroeder and co-authors. The study investigates the response of plant traits, particularly those related to plant reproduction and herbivory resistance, to increasing modifications in the agricultural landscape. Although the study is well-designed and up-to-date, with a wide range of measures obtained, there are still some issues that need to be addressed before publication. I have provided general comments here, while specific comments can be found in the attached document.

Introduction:

The introduction is too general and does not adequately prepare the reader for the rest of the manuscript. I suggest reducing the "General literature" section on the effects of agriculture on plant traits and phenotype and providing more detail on the impact on reproductive traits and herbivory resistance. Consider what the reader needs to be aware of before starting with the description of your experiment, rather than making general statements that do not clearly set the aims.

Material and Methods:

This section reflects the extensive effort put into the experiment, but I believe it is too long. Consider merging some sections, such as "Study system and landscape analysis," or removing details that are not essential to understanding the process.

Results:

I have no general comments to add here. As mentioned in the text, I suggest summarizing Figs. 3-7 into a single 5 x 3 matrix, with each figure as a column. This would greatly improve the interpretability of the work.

Discussion:

The Discussion section requires clarification. The study produced a lot of information on different plant traits, some influenced by agricultural landscape modification and others by other factors, with varying significance for each species. I recommend to clarify the general conclusions for the reader by creating a scheme or illustration that summarizes all the results and provides a quick overview. Additionally, consider adding a "Final Remarks" or "Conclusions" section at the end of the Discussion to summarize the most important results and those that are most likely to be useful to future researchers.

I hope these comments are helpful and inspiring and that your manuscript will be ready for publication soon. Good luck.

Reviewer #2: This study evaluates intraspecific variation of defense and reproductive traits in 3 plant species according to landscape intensification (increase in proportion of the landscape covered by agriculture).

The topic is very relevant, the experimental methods and analyses are sound and well executed, and the manuscript is clearly written. I feel, however, that the discussion section could be improved. The explanations of the observed patterns and possible mechanisms behind them, fall short after such well-performed study and carefully crafted introduction. I think providing more details about the collecting sites and the landscape profile and history would be very helpful for thinking about potential mechanisms (see below).

Besides that, I have only minor suggestions that may help improve the clarity of manuscript

Specifics:

Introduction

In general, very good and clear introduction.

Line 91: it’s not clear here if reproductive and defense traits were measured in the plants from the field margins or their progeny.

Methods:

In general, good descriptive section. I do think more details are needed for the collection sites.

Line 112: I think the authors need to provide a detailed description of the collection sites. It would help evaluate the methods as well as provide possible cues for understanding the results. I would like to have information about size and vegetation characteristics of the collecting sites. In addition, a map or a table of the distance between sites, the distribution of the sites along the gradient of landscape intensification, as well as types of agriculture in the buffers (most importantly for this last one, are there Brassica crop fields nearby?).

Line 117: why was a regression used to compare between years and not a correlation matrix? I think the latter would be more appropriate. Related to this, in general, I think there are too many figures. Fig. 1 is not really necessary; presenting the correlation values between years should be enough.

Line 127: please explain how plants where “chosen”

Line 128: this sentence is not clear “The natural variation in landscape composition at the broader scale from 500-1500m around each collection site represented a gradient with an

increasing proportion of agricultural land cover”. Do the authors mean that the sites fell on a landscape gradient? If so, I would like to see a distribution of the sites along this gradient (could be a figure in the supplemental materials or a table with the proportions of agriculture cover for each site).

Line 182: the sentence that starts at the end of this line is missing a “that”

Line 209: a PCA was run with “combining the most predictive landscape types for

each species across scales”. Please clarify what this means. How did they know in advance which where the ‘most predictive landscape types for each species’?

Line 220; figure 2: PC2 shows a clear gradient from “natural” to pastures. This gradient seems very relevant for the questions of the study. Why wasn’t it evaluated?

Line 237: Weren’t all plants for this species vernalized?

Line 242: sentence starting in this line is not clear.

Line 246: please briefly explain what ‘bolting status’ is and why was it important to include it as random effect.

Results

In general, the results are clearly presented but I do think the figures could be used more efficiently (see below)

256: please replace “higher agricultural landscapes” with a more descriptive term and use it consistently across the manuscript. The authors also use ‘landscape modification’, ‘increasing agricultural habitat modification’ and other terms to refer to the ‘increasing amount of agriculture cover in the landscape’. Use one consistent term across the manuscript.

261: These are regression coefficients (GLMMs) not correlation coefficients

Figures 3-7. Just a suggestion but I think these figures could be combined into one summary figure that would make it easier to see the found patterns. As they are now, it’s hard to contrast between the different reproductive and defense traits. The summary figure could just give information of the direction of the effects (+/-/0) between increasing ag. cover and each of the different traits (somehow separated by ‘defense’ ‘reproductive’). The detailed regression figures could be provided in the supplemental material.

Discussion:

In general, I think the discussion was the weakest section of the manuscript. I would suggest a more careful interpretation of the results. More information on the sites, on the natural history of the plants (and the plants surrounding them), on the history of the sites, may provide cues for the potential mechanism behind the results.

379: given that the authors are talking about selection pressures here, it would be worth it to, at the very least, mention how long these landscapes have been agriculture dominated (10 years may not be enough).

387: The explanation and interpretation for the results in this paragraph are not clear. It is not clear what the authors are concluding from these results: “caterpillars consumed less leaf area and developed more efficiently on plants from agriculturally dominant landscapes.” But given this sentence “Studies examining plant defenses in urban landscapes have found similar results to those presented here, demonstrating a decrease in plant defenses within modified urban landscapes (21,52)” tells me that the authors are saying that their results suggest decreased plant defenses in ag. dominated landscapes. I’m not sure why. Caterpillars consuming less leaf area and developing more efficiently means (or suggests) that the plants in ag. dominated areas are better defended, not less. This is not discussed in this paragraph.

6. PLOS authors have the option to publish the peer review history of their article (what does this mean?). If published, this will include your full peer review and any attached files.

Reviewer #1: No

Reviewer #2: No

---

## [Author Response · Author response to Decision Letter 0]

17 Apr 2023

1. I have reviewed the specific formatting guidelines and modified the manuscript as such. 

2. I have added a statement in the methods that no permits were required for this work.

3. I have provided an updated financial disclosure statement in the cover letter. This will match the funding statement.

4. The data are available in a dryad repository and is currently available for review here: https://datadryad.org/stash/share/bUYkXbBcTpQMpqADy88jskjmG1XOJerQIzI-1lTZT5o

---

## [Decision Letter · Decision Letter 1]

9 May 2023

Phenotypic clines in herbivore resistance and reproductive traits in wild plants along an agricultural gradient

PONE-D-23-01036R1

Dear Dr. Schroeder,

We’re pleased to inform you that your manuscript has been judged scientifically suitable for publication and will be formally accepted for publication once it meets all outstanding technical requirements.

Kind regards,

Andrea Mastinu

Academic Editor

PLOS ONE

Additional Editor Comments (optional):

Reviewers' comments:

Reviewer's Responses to Questions

**Comments to the Author**

1. If the authors have adequately addressed your comments raised in a previous round of review and you feel that this manuscript is now acceptable for publication, you may indicate that here to bypass the “Comments to the Author” section, enter your conflict of interest statement in the “Confidential to Editor” section, and submit your "Accept" recommendation.

Reviewer #1: All comments have been addressed

2. Is the manuscript technically sound, and do the data support the conclusions?

Reviewer #1: Yes

3. Has the statistical analysis been performed appropriately and rigorously? 

Reviewer #1: Yes

4. Have the authors made all data underlying the findings in their manuscript fully available?

Reviewer #1: Yes

5. Is the manuscript presented in an intelligible fashion and written in standard English?

Reviewer #1: Yes

6. Review Comments to the Author

Reviewer #1: All my suggestions have been adressed and I have no other concerns at this time.

Congratulations for your work.

7. PLOS authors have the option to publish the peer review history of their article (what does this mean?). If published, this will include your full peer review and any attached files.

Reviewer #1: No

---

## [Editor Report · Acceptance letter]

19 May 2023

PONE-D-23-01036R1 

Phenotypic clines in herbivore resistance and reproductive traits in wild plants along an agricultural gradient 

Dear Dr. Schroeder:

I'm pleased to inform you that your manuscript has been deemed suitable for publication in PLOS ONE. Congratulations! Your manuscript is now with our production department. 

Kind regards, 

on behalf of

Dr. Andrea Mastinu 

Academic Editor

PLOS ONE